# Efficient Multi-Agent System Training with Data Influence-Oriented Tree Search

## Abstract

Large Language Model (LLM) based multi-agent systems (MAS) show strong potential for tackling complex tasks through collaborative intelligence. Monte Carlo Tree Search (MCTS) based methods provide promising approaches for enhancing MAS self-training by generating synthetic data, using Q-values to estimate agent contributions. However, relying solely on Q-values may misalign with the goal of selecting data most beneficial for MAS improvement. To address this discrepancy, we propose **D**ata **I**nfluence-oriented **T**ree **S**earch (**DITS**), a novel framework that incorporates influence scores to guide both tree search and data selection in data synthesis. By leveraging influence scores, we effectively identify the most impactful data for MAS improvement, thereby enhancing model performance. Furthermore, we derive a novel influence score estimation method tailored for non-differentiable metrics, significantly reducing computational overhead by calculating performance changes on the validation set. Extensive experiments on three different multi-agent tasks demonstrate the robustness and effectiveness of the proposed methods. Notably, our findings reveal that allocating more inference resources to estimate influence scores, rather than Q-values, during data synthesis can more effectively and efficiently enhance model training. The code is available at https://anonymous.4open.science/r/DITS-F1C4/.

## 1 Introduction

LLM based agents have recently achieved remarkable success across a wide range of tasks (Hu et al., 2024; Wang et al., 2024b; Xi et al., 2023; Zhang et al., 2024a). Leveraging the advanced natural language understanding and reasoning capabilities of LLMs (OpenAI, 2023; Wei et al., 2022), these agents are able to dynamically interact with complex tools and environments to accomplish various tasks (Chen et al., 2023; Yao et al., 2023). Nevertheless, individual agents often face significant limitations when confronted with complex tasks (Shi et al., 2024b). In such scenarios, the multi-agent system (MAS) (e.g., MetaGPT (Hong et al., 2024), AutoGen (Wu et al., 2023), Camel (Li et al., 2023)) involving multiple specialized agents, with strategic task allocation and division of labor, becomes crucial for achieving optimal outcomes (Guo et al., 2024). However, optimizing the collective performance of LLM-based MAS as a cohesive unit and obtaining reward signals for each agent in the MAS still remain challenging problems (Chen et al., 2024b).

To tackle this challenge, leveraging synthetic data for self-training emerges as a highly promising direction. Monte Carlo Tree Search (MCTS) (Guan et al., 2025; Li et al., 2025a) based method offers a promising approach for synthetic data generation, capable of estimating individual agent contributions through Q-value (Chen et al., 2024b). They collect fine-grained preference pairs, encouraging high-Q-value actions while suppressing low-Q-value actions via Direct Preference Optimization (DPO) (Rafailov et al., 2023). Despite its potential, the current tree search strategy is primarily adapted from the inference phase, inheriting its inherent characteristics, which rely on Q-values to identify informative data. This reliance misaligns with the data synthesis objective, which focuses on generating data that better facilitates model training. The empirical results presented in Figure 1 (a) (b) (c) also demonstrate that actions associated with higher Q-values do not always contribute significantly to the improvement of model performance, where the influence score serves as a metric to quantify the utility of data in enhancing model performance.

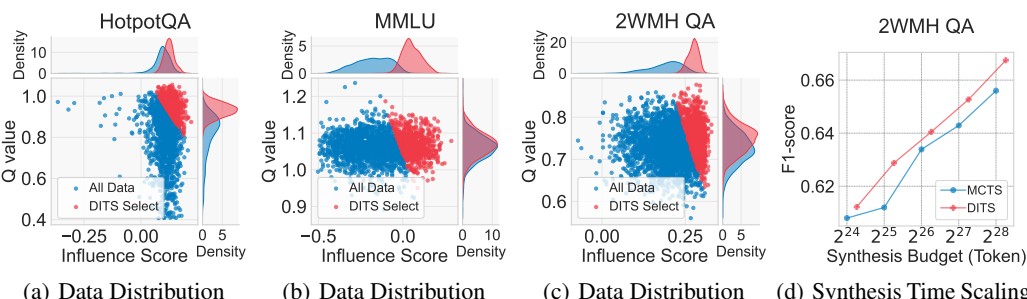

(a) Data Distribution  (b) Data Distribution  (c) Data Distribution  (d) Synthesis Time Scaling

Figure 1: (a) (b) (c) The scatter plot and density plots of Q-values and influence scores for synthetic data. The top 30% of the data selected using DITS is highlighted in red. (d) Performance trends with different data synthesis budgets (Tokens).

To address this issue, we propose **D**ata **I**nfluence-oriented **T**ree **S**earch (**DITS**), a novel framework that optimizes MAS through iterative synthetic data generation guided by influence-aware tree search. Our approach combines MCTS for MAS trajectory simulation with a data influence mechanism that prioritizes training samples based on their expected contribution to model improvement, rather than relying solely on traditional Q-value estimates. The influence score quantifies how training data impacts model outputs, helping identify data points that most improve performance. While traditional methods rely on training loss as a performance metric, this is less effective for DPO loss due to its weak correlation with downstream performance (Rafailov et al., 2024; Shi et al., 2024c). Hence, we redefine the influence score based on the changes in non-differentiable metrics on the validation set and derive a novel estimation method. Our method circumvents computationally intensive gradient computations across large-scale parameters that are required in traditional approaches.

We validate our approach on seven datasets across three multi-agent tasks: Information Exchange, Debate (Chen et al., 2024b), and DeepSearch (Li et al., 2025c). We observe that high Q-value data may reduce the diversity of the model's responses and contribute little to improving model performance. Incorporating data influence is crucial for data synthesis and selection. Our method outperforms state-of-the-art multi-agent optimization techniques, achieving an average improvement of 2.7% in single-round iterations, a 2.5% performance enhancement in multi-round iterations for the Information Exchange task, and 2.6% performance improvement for the DeepSearch task. Within the same data synthesis budget, our method surpasses traditional approaches, delivering more efficient scaling of synthesis computation, as shown in Figure 1 (d) and in Appendix B.

We summarize the main contributions as follows:

- We propose DITS, a novel framework that employs influence scores to guide tree search and data selection. This enables the prioritized selection of preference pairs that contribute more significantly to performance improvement.

- We derive the influence score estimation method for non-differentiable metrics. This approach substantially reduces computational overhead through inference computation, enabling more efficient synthesis time scaling.

- We achieve state-of-the-art performance across multiple multi-agent tasks and demonstrate that the framework's capability can be continuously improved through iterative rounds of data synthesis.

## 2 RELATED WORK

LLM-based multi-agent systems (MAS) have demonstrated remarkable capabilities in addressing complex problems in various tasks (Hong et al., 2024; Islam et al., 2024; Tran et al., 2025). These systems employ various collaborative strategies, including multi-agent debate (Du et al., 2024; Liang et al., 2024) and role-based division of labor (Qian et al., 2024a; Wang et al., 2024d). Researchers have explored several key approaches to improve the performance of multi-agent systems. One strategy focuses on expanding the diversity and scale of agents (Li et al., 2024a; Qian et al., 2024b; Wang et al., 2024a), optimizing performance from a network architecture perspective. Another approach emphasizes enhancing prompt quality, such as refining system memory in frameworks

like AutoGen (Wu et al., 2023) and BiLLP (Shi et al., 2024a) or improving instruction design and few-shot examples in Dspy (Khattab et al., 2023; Opsahl-Ong et al., 2024). A third approach involves fine-tuning the parameters of the large models within the agents, which is the most effective yet challenging method. Optima (Chen et al., 2024b) and MALT (Motwani et al., 2024) have taken the first step in this direction by constructing preference training data pairs through estimating Q-values. MALT can be viewed as a special case of Optima.

MCTS is an advanced search algorithm capable of effectively balancing exploration and exploitation in decision-making processes. It gained significant attention following its success in AlphaGo (Silver et al., 2016). Subsequently, researchers have introduced MCTS into LLM reasoning tasks (Hao et al., 2023), giving rise to two primary methodologies. The first approaches employ MCTS during the inference phase, prioritizing actions with the highest potential to yield correct outcomes (Snell et al., 2024; Wu et al., 2024). The second approaches leverage MCTS during the training phase to synthesize high-quality training data, with the goal of identifying data that maximizes the improvement in model performance (Qi et al., 2024; Xie et al., 2024; Zhang et al., 2024c;b). These approaches mainly rely on estimated Q-values to guide the exploration of the synthesis data space.

The influence function, first introduced by Hampel (1974), assesses the impact of individual data points on model performance and has become a powerful tool for training data valuation. Unlike alternative approaches such as LLM-based rating methods (Liu et al., 2024) or reward function methods (Wang et al., 2024c), the influence function offers distinct advantages by quantifying data utility through rigorous mathematical analysis of model training dynamics. Recent studies have extended its use to improve data quality in LLM pre-training through TraceIn (Pruthi et al., 2020) and MATES (Yu et al., 2024), for instruction tuning with Montessori-instruct (Li et al., 2024b) and LESS (Xia et al., 2024), and for reward modeling with OPORP (Min et al., 2025). However, its potential for MAS data synthesis that maximizes system capability enhancement remains unexplored. The core challenge in applying influence functions lies in its high computational cost. Classical methods, such as gradient-based approaches (Koh & Liang, 2017; Park et al., 2023) and trajectory-influence based methods (Bae et al., 2024), require the computation of billion-level gradients, which is extremely expensive. For efficient estimation, MATES (Yu et al., 2024) probes the oracle data influence by evaluating the model's reference loss after training on individual data points. Our approach extends the reference loss to non-differentiable validation metrics, thereby enabling the enhancement of data quality through data synthesis.

## 3 METHOD

In this section, we first formalize the multi-agent task and MCTS-based data synthesis (§ 3.1), then introduce the data influence-oriented data selection (§ 3.2), and finally present the iterative data synthesis process (§ 3.3).

### 3.1 MULTI-AGENT TRAINING DATA SYNTHESIS

Training effective MAS requires high-quality data that reflects complex agent interactions, but collecting such data in the real world is costly and time-consuming. To overcome this, we utilize MCTS to simulate interactions and automatically produce preference-labeled training data.

In this work, we model the topology structure for multi-agent collaboration as a directed graph. Concretely, we denote a feasible topology as $\mathcal{G} = (\mathcal{V}, \mathcal{E})$, as demonstrated in Figure 2 (a). It is worth noting that such graph structures can be static or dynamic, with the dynamic variant allowing agents to govern the information flow in an adaptive manner. We allow the presence of cycles in the graph, indicating that multiple rounds of information exchange are permitted among agents $A$. We assume that our agent network can be linearly traversed in topological order $A_1 \oplus A_2 \oplus \cdots \oplus A_M$ Bondy & Murty (1976); Gross & Yellen (2005); Qian et al. (2024c), where $A_m \in \mathcal{V}$. Different $A_m$ may represent the same agent being visited at different time steps. For clarity and convenience, we use different symbols to distinguish them.

In this way, we could utilize MCTS to synthesize training data for MAS. We mainly follow the configuration in Optima Chen et al. (2024b) and construct the tree as follows: As shown in Figure 2 (b), the synthesis tree begins with a specific task instruction $p$.

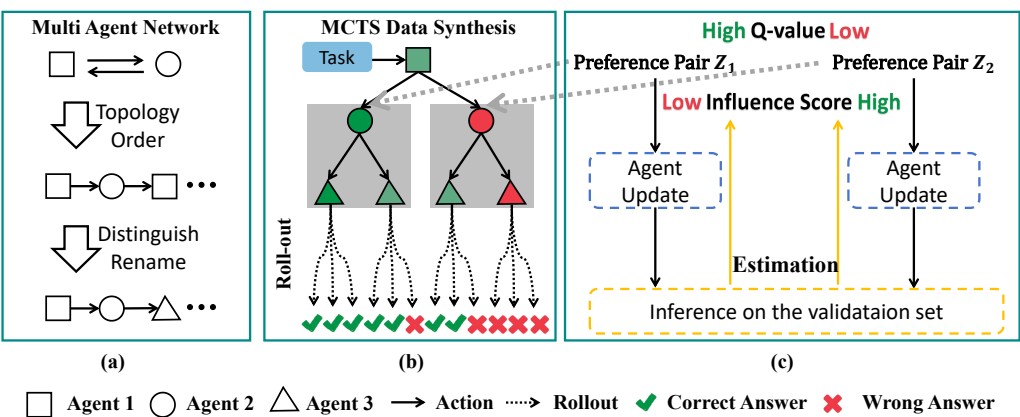

Figure 2: Overview of our method. (a) illustrates the traversal of a cyclic agent network in topological order. We introduce virtual agents to distinguish the same agent in the traversal. (b) showcases the application of MCTS to generate synthetic multi-agent training data, where the color of each agent represents the magnitude of the node's Q-value. (c) depicts the computation process of influence scores for a non-differentiable metric, highlighting that data points with high Q-values may correspond to low influence scores.

**Selection**: We select a node $n$ to expand from the candidate node set, where a node $n = (s, a)$ refers to an agent $A_m$ in state $s$ that takes action $a$. We use the edit distance to filter out nodes that are similar to expanded nodes to obtain the candidate node set.

$$N_{\text{cand}} = \{n_j | n_i \in N_{\text{expanded}}, n_j \in N_{\text{all}}, S_{i,j} \geq 0.25\}, \tag{1}$$

where $S_{i,j} = \frac{\text{edit\_distance}(n_i, n_j)}{\max(|n_i|, |n_j|)}$ and edit_distance$(n_i, n_j)$ represents the edit distance between the action strings of two nodes. $N_{\text{all}}$ and $N_{\text{expanded}}$ denotes the whole node set and expanded node set. Then we select a node for the candidate set $N_{\text{cand}}$ based on softmax distribution of Q-values.

$$n \sim \text{Softmax}(\{Q(n)\}_{n \in N_{\text{cand}}}), \tag{2}$$

where $Q(n) = Q(s, a)$ and the softmax distribution balances exploration and exploitation.

**Expansion** For each selected node $n$, we denote the new state as $s' = \text{Trans}(s, a)$, where $\text{Trans}(\cdot)$ is the transit function determined by the environment. Then we sample $d$ actions from agents $A_{m+1}$:

$$\{a'_1, \cdots, a'_d\} \sim A_{m+1}(s'). \tag{3}$$

**Simulation** For each generated action $a'_i$, we simulate the agent interaction $\tau_i$ until termination.

$$\tau_i = \text{Simulation}(A_{m+2}, \cdots, A_M, s', a'_i). \tag{4}$$

Meanwhile, we construct all $(s, a)$ pairs in the trajectory as new nodes and add them to $N_{\text{all}}$.

**Backpropagation** Once a trajectory $\tau$ is completed, we can obtain the trajectory reward $R(\tau)$ detailed in Appendix D. We update the Q-value of nodes with the average rewards from the trajectories set containing the node.

$$Q(n) = Q(s, a) = \sum_{\tau \in \mathcal{T}(n)} \frac{1}{|\mathcal{T}(n)|} R(\tau), \tag{5}$$

where $\mathcal{T}(n)$ denotes the trajectory set containing the node $n$. Additionally, due to the complex interactions among multiple agents, the Q-value estimates obtained from $d$ rollouts may be inaccurate. Allocating more inference budget in the data synthesis phase may improve the quality of the generated data and enhance the system's performance.

We repeat the above process $k$ times and finish the generation process. Then we can construct paired action preferences for agent $A_i$ at state $s$ by selecting the action $a_i^h$ with the highest Q-value and the action $a_i^l$ with the lowest Q-value to form the preference data:

$$z = \left(s, a_i^h, a_i^l\right). \tag{6}$$

To update the parameter of agent $A_i$, we utilize the Direct Preference Optimization (DPO) loss to encourage the model to prioritize responses that align with preferences $a_i^h$ over less preferred ones $a_i^l$.

$$\mathcal{L}_{DPO} = \mathbb{E}_z \left[ -\log \sigma \left( \beta \left[ \log \frac{\pi_\theta(a_i^h \mid s)}{\pi_{\text{ref}}(a_i^h \mid s)} - \log \frac{\pi_\theta(a_i^l \mid s)}{\pi_{\text{ref}}(a_i^l \mid s)} \right] \right) \right], \tag{7}$$

where $\sigma(\cdot)$ denotes the sigmoid function and $\pi_{\text{ref}}$ represents the reference model, *i.e.* the SFT model.

## 3.2 DATA INFLUENCE-ORIENTED DATA SELECTION

While improving the accuracy of Q-value estimation can enhance data quality to some extent, it is both highly inefficient and suboptimal. During the training phase, the primary goal of synthetic data is to maximize its contribution to model performance improvement, rather than ensuring the data is correct. Figure 2 (c) reveals an important insight: while the data pair $z_1$ achieves higher Q-values, the data pair $z_2$ demonstrates greater practical impact on system performance. This suggests that absolute Q-values may not fully capture data pair's true contribution.

Hence, in this paper, we introduce the influence score $\mathcal{I}$ to quantify the impact of data on the current agent's performance. The influence score $\mathcal{I}$ was developed to measure the difference in loss when a data point is assigned a higher weight in the training dataset. Suppose the agent $A$ is parameterized by $\theta$. We denote the optimal parameters learned by minimizing the training loss $\mathcal{L}_{\text{tr}}$ on the dataset $\mathcal{D}_{\text{tr}}$, with a data point $z_i$ assigned an additional weight of $\epsilon$, as:

$$\theta_{\epsilon, z_i}^* = \arg\min_\theta \sum_{z_j \in \mathcal{D}_{\text{tr}}} \frac{1}{|\mathcal{D}_{\text{tr}}|} \mathcal{L}_{\text{tr}}(z_j, \theta) + \epsilon \mathcal{L}_{\text{tr}}(z_i, \theta). \tag{8}$$

Under standard assumptions, such as the twice-differentiability and strong convexity of the loss function $\mathcal{L}_{\text{tr}}$, the influence function can be derived via the chain rule of the derivatives (Koh & Liang, 2017). However, the DPO loss does not effectively align with downstream task performance. Our experiments reveal a weak correlation (less than 0.2) between the DPO loss and performance metrics $\mathcal{F}$ such as F1-score or Accuracy on the validation set. This observation is consistent with findings reported in (Rafailov et al., 2024; Shi et al., 2024c). This indicates that we must redefine the influence score using the changes of non-differentiable performance metrics on the validation set.

$$\mathcal{I}_{\mathcal{F}_{\text{val}}}(z_i, \mathcal{D}_{\text{val}}) := \frac{\mathcal{F}_{\text{val}}(z_i, \theta_{\epsilon, z_i}^*) - \mathcal{F}_{\text{val}}(z_i, \theta^*)}{\epsilon}, \tag{9}$$

where $\theta^* = \theta_{\epsilon, z_i}^* |_{\epsilon=0}$. Due to non-differentiable metric $\mathcal{F}_{\text{val}}$, the influence function cannot be derived using gradients. Instead, we use the finite difference method combined with parameter perturbation to approximate the rate of change. The perturbed optimal parameter $\theta_{\epsilon, z_i}^*$ can be rewritten as:

$$\theta_{\epsilon, z_i}^* = \theta^* + \epsilon \Delta\theta + o(\epsilon), \tag{10}$$

where $\Delta\theta$ represents the direction of parameter change. Following Yu et al. (2024), the direct is typically driven by the gradient of the training loss.

$$\Delta\theta \propto -\nabla_\theta \mathcal{L}_{\text{tr}}(z_i, \theta^*). \tag{11}$$

Since the parameter update is dominated by the training loss gradient, we adopt a one-step gradient descent update:

$$\theta_{\epsilon, z_i}^* \approx \theta^* - \eta\epsilon \nabla_\theta \mathcal{L}_{\text{tr}}(z_i, \theta^*), \tag{12}$$

where $\eta$ is the learning rate, and $\epsilon$ is a very small perturbation strength. Combining the finite difference and parameter update, the influence function is approximated as:

$$\mathcal{I}_{\mathcal{F}_{\text{val}}}(z_i, \mathcal{D}_{\text{val}}, \theta^*) \approx \frac{1}{\epsilon} \left[ \mathcal{F}_{\text{val}}(z_i, \theta^* - \eta\epsilon \nabla_\theta L_{\text{tr}}(z_i, \theta^*)) - \mathcal{F}_{\text{val}}(z_i, \theta^*) \right]. \tag{13}$$

Following Koh & Liang (2017), we theoretically illustrate that selecting data points with the highest influence scores maximizes the model's validation performance (see Appendix A for details). Finally, our selection strategy combines Q-values and influence scores to effectively identify the highest-quality pair data:

$$H(z_i) = \mathcal{I}_{\mathcal{F}_{\text{val}}}(z_i, \mathcal{D}_{\text{val}}, \theta) + \gamma \cdot Q(s, a_i^h), \tag{14}$$

where $\theta$ denotes the current parameters of agent $A_m$. Finally, after filtering out low-quality data as described in Chen et al. (2024b), synthetic data are ranked based on the scores, and the Top $\alpha$ are selected to construct the training dataset $\mathcal{D}_{\text{tr}}$.

Table 1: **Performance comparison across Information Exchange and Debate tasks.** Best results are indicated in **bold**, and second-best results are underlined. The baseline results are taken from Chen et al. (2024b).

| | Information Exchange | | | | Debate | |
|---|---|---|---|---|---|---|
| Method | HotpotQA | 2WMH QA | TriviaQA | CBT | ARC-C | MMLU |
| CoT | 25.6 | 20.5 | 59.8 | 43.4 | 65.2 | 46.0 |
| MAD | 28.4 | 25.9 | 71.0 | 53.8 | 71.4 | 51.5 |
| AutoForm | 28.2 | 24.7 | 60.9 | 35.0 | 60.2 | 43.8 |
| Optima-iSFT | 54.5 | 72.4 | 71.9 | 71.8 | 74.1 | 56.8 |
| Optima-iDPO | 52.5 | 66.1 | 69.3 | 66.7 | 74.5 | 59.6 |
| Optima-iSFT-DPO | 55.6 | 74.2 | 77.1 | 70.1 | 77.1 | 60.2 |
| DITS-iSFT-DPO | **57.2** | **76.0** | **78.4** | **72.0** | **77.6** | **60.5** |

## 3.3 ITERATIVE DATA SYNTHESIS

In addition to utilizing the current model for data synthesis, we propose an iterative refinement approach to generate higher-quality data. By continuously training and enhancing the model, its capabilities improve, enabling the generation of more valuable synthetic data in subsequent iterations. At iteration $t$, we generate the training dataset $\mathcal{D}_{\text{tr}}^t$ based on the parameters $\theta_{t-1}$ and train a new model from the initial model using $\mathcal{D}_{\text{tr}}^t$. The corresponding pseudocode can be found in Algorithm 1.

## 4 EXPERIMENTAL SETUP

In this section, we will introduce the datasets, metrics, and baseline methods in the experiments.

**Dataset** To validate the collaborative and task allocation capabilities of MAS, we evaluate our framework DITS in three multi-agent settings: two static scenarios—**Information Exchange** and **Debate**—and one dynamic scenario, **DeepSearch**. In the information exchange setting, the relevant context is divided between two agents. The agents must identify the relevant information and communicate with each other to derive the final answer. This setting includes HotpotQA (Yang et al., 2018), 2WikiMultiHopQA (2WMH QA) (Ho et al., 2020), TrivalQA (Joshi et al., 2017), and CBT (Hill et al., 2016). In the debate setting, two agents work together to solve a task: one agent proposes solutions, while the other evaluates their correctness. The debate setting includes ARC's challenge set (ARC-C) (Bhakthavatsalam et al., 2021) and MMLU (Hendrycks et al., 2021). Unlike static scenarios, where multi-agent collaboration follows a predetermined sequence, the dynamic DeepSearch setting features agents that autonomously determine and continuously adjust their collaboration strategies based on the evolving task, enabling truly adaptive and intelligent teamwork. The DeepSearch task involves collaboration among a task analysis agent, a search intent generation agent, and a web content analysis agent. The DeepSearch setting includes WebWalker (Wu et al., 2025). We use 0-shot for all benchmarks.

**Metrics** Following Chen et al. (2024b), we employ the F1 score between final answers and labels as evaluation metrics for information exchange tasks. For debate tasks, we utilize exact match accuracy (ARC-C, MMLU). For the deepsearch task, following Wu et al. (2025), we utilize Qwen2.5-72B-Instruct (Team, 2025) to verify whether the answers were consistent with the correct answers.

**Baseline** For static scenarios, we compare our methods with: (1) Chain-of-Thought (CoT) (Wei et al., 2022): single agent pipeline which enables complex reasoning to derive the final answer. (2) Multi-Agent Debate (MAD) (Du et al., 2024): multi-agent pipeline where different reasoning processes are discussed multiple rounds to arrive at the final answer. (3) AutoForm (Chen et al., 2024a): multi-agent pipeline where the agents utilize non-nature language formats in communication to improve efficiency. For the dynamic scenario, following Li et al. (2025c), we evaluate several pipeline methods within this setting, including (1) Direct Reasoning, (2) RAG workflow and its variant (Li et al., 2025c), and (3) Search-o1 (Li et al., 2025b). In both scenarios, we compare DITS with multi-agent optimization method Optima (Chen et al., 2024b): a framework that enhances communication efficiency and task

effectiveness through Supervised Finetuning and Direct Preference Optimization. It has three variants, namely Optima-iSFT, Optima-iDPO, and Optima-iSFT-DPO. We follow the iSFT-DPO variant of Optima and improve its data synthesis and selection process to obtain DITS-iSFT-DPO.

**Implementation Details** We utilize the Llama-3-8B-Instruct (Dubey et al., 2024) as the base model for static scenarios. Experimental results for other base models are provided in Appendix G.4. For the dynamic scenario, we employ the QwQ-32B (Team, 2024) as the base model due to the task complexity. The interaction ends when either a special token marks the final answer or the maximum number of turns is reached. Unless otherwise specified, we set the hyperparameters to $\alpha = 0.5$ and $\gamma = 1$. When collecting influence scores via single-step gradient descent, we utilize LoRA (Low-Rank Adaptation) (Hu et al., 2022). A validation set of size 20 is used in all experimental settings. We set the expansion time $d = 3$ and repeat time $k = 8$ for all datasets. More details are provided in the Appendix E.

## 5 EVALUATION RESULTS

In this section, we first evaluate the effectiveness of DITS (§ 5.1). Then we demonstrate the superiority of data influence through ablation studies (§ 5.2) and explore the impact of synthesis scaling on data quality (§ 5.3). Finally, we analyze the effects of selection ratio and iteration times (§ 5.4).

### 5.1 OVERALL PERFORMANCE

**The Static Scenarios** In Table 1, we compare our method DITS-iSFT-DPO with the baseline approaches on both the Information Exchange and Debate tasks. Across all datasets, our method achieves consistent improvement over the baselines, demonstrating the effectiveness and generalizability of DITS. Compared to the single agent CoT approach, our method delivers an average performance enhancement of 91%. In the Information Exchange task, our method outperforms the advanced multi-agent approach Optima-iSFT-DPO by an average margin of 2.5%.

**The Dynamic Scenario** In dynamic scenarios, we adopt the WebThinker framework (Li et al., 2025c) to structure the process into a collaborative system comprising three agents: a task analysis agent, a search intent generation agent, and a web content analysis agent. This framework empowers the agents to autonomously conduct web searches, deeply analyze web content, and dynamically adjust their collaboration strategies. For search, we use the Serper API[1], retrieving the top 10 search results (k=10). In Table 2, we observe that the Webthinker framework for dynamic multi-agent collaboration outperforms traditional single-agent approaches and simple RAG methods. Furthermore, fine-tuning multiple agents within the collaborative framework enhances coordination efficiency. Notably, the DITS method surpasses all baseline models, highlighting its effectiveness and robustness.

Table 2: Performance comparison on DeepSearch task. Best results are indicated in **bold**.

| Models | WebWalker |
|---|---|
| Direct Reasoning | 4.3 |
| RAG Workflow | 31.2 |
|   - w/ Query Planning | 32.5 |
|   - w/ Iterative RAG | 31.5 |
| Search-o1 | 34.1 |
| WebThinker | |
|   - Base | 37.0 |
|   - Optima-SFT | 46.0 |
|   - Optima-DPO | 46.6 |
|   - DITS-DPO | **47.2** |

### 5.2 INFLUENCE FUNCTION ANALYSIS

To provide a detailed comparison of the effectiveness of the influence function, we present the results of different data selection methods in Table 3. The experiments are conducted in a single iteration. The Base method represents the multi-agent framework performance with the base model Llama-3-8B-Instruct. The Optima-DPO and Optima-RPO methods utilize the dataset $\mathcal{D}_{tr}$ sampled through the MCTS approach in Optima to train the model using DPO loss (Rafailov et al., 2023) and RPO loss (Pang et al., 2024), respectively. Random Select refers to training on the data randomly sampled from $\mathcal{D}_{tr}$ with DPO loss, while Q-value Select involves selecting the top-ranked data based

---

[1] https://serper.dev/

Table 3: **Single iteration performances across Information exchange and Debate tasks.** Best results are indicated in **bold**, and second-best results are underlined.

| Method | Information Exchange | | | | Debate | |
|---|---|---|---|---|---|---|
| | HotpotQA | 2WMH QA | TriviaQA | CBT | ARC-C | MMLU |
| Base | 28.2 | 24.7 | 60.9 | 35.0 | 60.2 | 43.8 |
| Optima-SFT | 45.2 | 59.7 | 68.8 | 50.7 | 68.2 | 50.3 |
| Optima-RPO | 50.4 | 60.6 | 68.4 | 59.1 | 72.2 | 52.1 |
| Optima-DPO | 46.6 | 61.2 | 70.9 | 57.2 | 71.5 | 51.6 |
|   - Random Select | 51.5 | 60.6 | 70.3 | 58.0 | 74.0 | 51.1 |
|   - Q-value Select | 50.5 | 61.1 | 69.8 | 58.6 | 73.7 | 50.2 |
| DITS-DPO | | | | | | |
|   - $\gamma = 0$ | **53.1** | **62.2** | **72.2** | **59.6** | 74.2 | 50.8 |
|   - $\gamma = 1$ | 52.8 | 61.5 | 71.0 | 59.1 | **74.5** | **52.3** |

on Q-values for training. DITS employs the influence score in Eq. equation 14 to select the top-ranked data for training, where the variant $\gamma = 1$ integrates both Q-value and influence score, and the variant $\gamma = 0$ relies solely on the influence score for data selection. For a fair comparison, we set the selection ratio as 50% for all methods.

**Ablation Study** As shown in Table 3, we observe that (1) The DITS method achieves consistent performance improvements across all datasets compared to using the full dataset, indicating that the original MCTS-generated dataset contains noisy and lower-quality data. This suggests that further enhancing data quality is beneficial for model performance. (2) Selecting data based on influence scores outperforms both random selection and Q-value-based selection, highlighting its superior effectiveness in enhancing data quality. To further explore the underlying reasons for this improvement, the following paragraph provides an in-depth analysis of the data distribution. (3) For the Information Exchange task, the variant $\gamma = 0$ achieves the best performance, while the variant $\gamma = 1$ achieves suboptimal results. In contrast, on the Debate task, the variant $\gamma = 1$ generally performs the best. This discrepancy is attributed to the fact that the evaluation metric for the Information Exchange task is F1-score, which introduces more noise into the estimated Q-values, resulting in lower quality in selecting data.

**Distribution Analysis** To provide an in-depth analysis of the advantages of using the influence score for data selection, we visualize the distributions of Q-values and influence scores on the HotpotQA, MMLU, and 2WMH QA datasets in Figure 1 (a) (b) (c), highlighting the distribution of the top 30% data points selected by our methods with $\gamma = 1$. The visualization results of other datasets can be found in Figure 4. From the figures, we observe that: (1) There are discrepancies between the influence score and Q-value, which reveals that Q value is not perfectly aligned with training needs. This highlights the importance of integrating the influence score into the MCTS process and data selection process. (2) The data selected by our methods exhibit high influence scores and Q-values, indicating that DITS is capable of selecting high-quality data.

## 5.3 SYNTHESIS TIME SCALING

In this study, we empirically demonstrate that increasing the synthesis budget during the data synthesis phase enhances model performance, as shown in Figure 1 (d) and Appendix B. Specifically, the figure highlights three key observations: (1) Allocating a larger synthesis budget, which extends rollout times and increases the number of expansions, will generate more high-quality data, thereby improving model performance. (2) We validate that allocating resources to influence score estimation can indeed lead to better performance improvements. This is attributed to the fact that the influence score is more aligned with training needs. This underscores the capability of our method to enhance the efficiency of synthesizing training data within a vast action space. (3) The performance gains from a sixteenfold increase in the synthesis budget are notably smaller compared to the improvements achieved through three times iterative data synthesis and training, as detailed in Table 1. This comparison highlights the efficiency and effectiveness of the iterative approach.

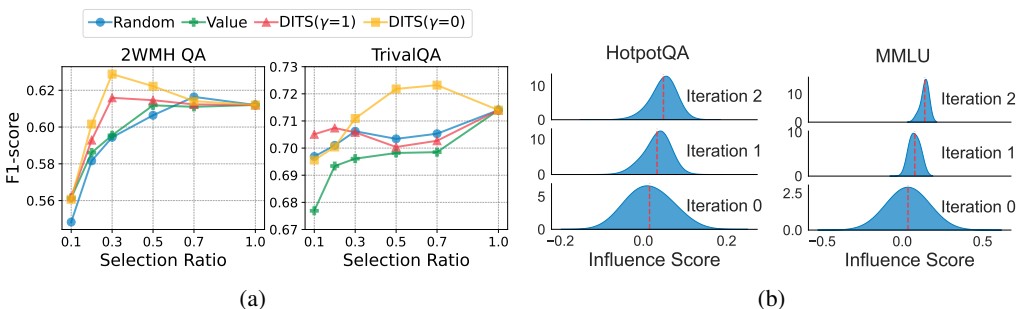

Figure 3: (a) The effect of hyperparameter selection ratio $\alpha$ of DITS on the 2WMH QA and TrivalQA datasets. (b) The distribution of synthetic data influence scores across different iterations on the HotpotQA and MMLU datasets, with the mean of the distribution highlighted by a red dashed line.

We also compare DITS with traditional data influence estimation methods. Unlike conventional gradient-based approaches, DITS offers improved computational efficiency. Additional details are provided in Appendix B.

### 5.4 Hyperparameter Analysis

**Selection Ratio** We first investigate the impact of the selection ratio hyperparameter $\gamma$ on model performance. We conduct experiments on two Information Exchange tasks: 2WMH QA and Trival QA datasets and present the results in Figure 3. We compare Optima-DPO (random Select and Q-value Select) with DITS ($\gamma = 0$) and DITS ($\gamma = 1$). From the figure, we observe that: (1) Across different selection ratios, DITS consistently outperforms Optima-DPO, demonstrating that our method can select data more beneficial for model training and exhibits strong generalization ability. (2) When an appropriate selection ratio is chosen, the performance of DITS surpasses that of using the full dataset, indicating the presence of noise in original MCTS-generated data and the potential for further improving data quality. (3) When the selection ratio is very small, the performance of all methods declines, indicating that training set size is also crucial for achieving optimal performance. This suggests that an overly small yet high-quality dataset may not be sufficient to train a good model.

**Iteration Times** To gain deeper insights into the iterative data synthesis and training process, we analyzed the distribution of influence scores for synthetic data across different iterations on the HotpotQA and MMLU datasets, as shown in Figure 3 (b). The mean of each distribution is highlighted. From the figure, we observe the following trends: (1) As the number of iterations increases, the mean influence score gradually rises, indicating an improvement in the quality of synthetic data. This suggests that the iterative process enhances data quality by refining the model, creating a positive feedback loop that makes data synthesis more effective. (2) With more iterations, the distribution of influence scores becomes more concentrated, suggesting that the model trained on synthetic data achieves more stable quality on specialized tasks. However, this may come at the cost of reduced data diversity.

We further analyze model performance over training iterations, the impact of validation set size, and compare data selection strategies based on influence scores. Details are provided in Appendix G.

## 6 Conclusion

In this work, we propose DITS, a novel multi-agent data self-training framework that integrates influence scores into MCTS to guide tree search and data selection. By leveraging influence scores and proposing a novel estimation method, we effectively identify the most impactful data for system improvement, thereby enhancing model performance. Meanwhile, we derive an efficient influence score estimation method for non-differentiable metrics through gradient-to-inference conversion. This approach substantially reduces computational overhead through inference computation and allows us to estimate influence scores to achieve a more efficient data synthesis process. Our approach introduces new perspectives and scaling dimensions for data synthesis, highlighting the impact of training data on model performance rather than its correctness.

## 7 REPRODUCIBILITY STATEMENT

To ensure the reproducibility of our work, we have taken several steps to provide comprehensive details on our methods, experiments, and implementations. First, an anonymized link to the source code has been included as supplementary material, allowing full access to our implementation. Second, section 3 elaborates on the key ideas and procedural details of our approach. Additionally, a comprehensive description of all hyperparameter configurations employed in the experiments is provided in the Section 4 and in Appendix D. We hope these resources will facilitate the replication of our results and further research.

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

Table 4: Comparison of training costs and performance between DITS and Optima on the 2WMH QA dataset.

| Method | Synthesis Budget (Token) | # Samples | Synthesis Cost (GPU Hours) | Training Cost (GPU Hours) | Total Cost (GPU Hours) | Performance (F1 score) |
|---|---|---|---|---|---|---|
| Optima-DPO | $1.67 \times 10^7$ | 17000 | 82 | 16 | 98 | 0.607 |
| Optima-DPO | $3.34 \times 10^7$ | 34000 | 165 | 30 | 195 | 0.610 |
| DITS-DPO | $2.00 \times 10^7$ | 8500 | 98 | 8 | 106 | 0.612 |

## A  THEORETICAL ANALYSIS

In this section, we illustrate the relationship between influence scores and model performance, where the selection of the most influential data points maximizes the model's validation performance. Concretely, we first extend the definition of influence score to a data group $U \subset \mathcal{D}_{tr}$ as:

$$\theta^*_{\epsilon,U} = \arg\min_\theta \sum_{z \in \mathcal{D}_{\mathrm{tr}}} \frac{1}{|\mathcal{D}_{\mathrm{tr}}|} \mathcal{L}_{\mathrm{tr}}(z,\theta) + \epsilon \sum_{z \in U} \mathcal{L}_{\mathrm{tr}}(z,\theta).$$

Following Koh & Liang (2017), under the first-order approximation, we have

$$\mathcal{I}_{\mathcal{L}_{\mathrm{tr}}}(U, \mathcal{D}_{\mathrm{tr}}) \stackrel{\text{def}}{=} \frac{d\mathcal{L}_{\mathrm{tr}}(U, \theta^*_{\epsilon,U})}{d\epsilon}\bigg|_{\epsilon=0} \approx \sum_{z \in U} \mathcal{I}_{\mathcal{L}_{\mathrm{tr}}}(z, \mathcal{D}_{\mathrm{tr}}),$$

where the influence score of a group of data points can be represented as the sum of the influence score of individual data points. For DITS, we adopt a similar approximation:

$$\mathcal{I}_{\mathcal{F}_{\mathrm{val}}}(U, \mathcal{F}_{\mathrm{val}}) \approx \sum_{z \in U} \mathcal{I}_{\mathcal{F}_{\mathrm{val}}}(z, \mathcal{D}_{\mathrm{val}}).$$

Thus, the selection of the most influential data points maximizes validation performance.

## B  EFFICIENCY ANALYSIS

In this section, we first provide an empirical computation cost comparison between DITS and Optima. Using the 2WMH QA dataset as an illustrative example, we compare the training costs per iteration across different settings. As shown in Table 4, we can observe that although employing data influence incurs additional costs, we argue that estimating the influence score is more effective in enhancing model performance compared to Optima.

Moreover, we quantify the computational cost of the different influence score estimation methods. For the forward pass of LLM, the computational cost is $2NS + 4LHS^2$, where $S$ is the sequence length, $N$ is the number of model parameters, $L$ is the number of model layers, and $H$ is the embedding dimension of the model. For small $S$ (e.g., 2000), the second term is negligible, making the cost per token $2N$. The backward pass doubles this cost to $4N$. During inference with KV cache, generating one token also costs $2N$.

We compare two classic gradient-based data influence methods, TRAK (Park et al., 2023) and LESS (Xia et al., 2024), under the following assumptions: average sequence length $S = 2000$, validation set size $V = 20$, model parameters $N = 8B$, projection dimension, $d = 8192$ for TRAK and LESS, and $R = 4$ checkpoints for LESS.

The computational costs (in FLOPs) for estimating the influence score of one data point are described in Table 5. As shown in the table, our method exhibits superior efficiency compared to traditional approaches. This advantage primarily stems from the fact that gradient-based methods require gradient computation on validation data and necessitate parameter projection onto a low-dimensional subspace.

Table 5: The computational costs comparison for estimating the influence score of one data point for different methods.

| Methods | FLOPs | FLOPs $(10^{15})$ |
|---------|-------|-------------------|
| DITS | $6N \cdot S + 2N \cdot V \cdot S$ | 0.7 |
| TRAK | $6N \cdot S \cdot V + 2N \cdot V \cdot d + V \cdot d^3$ | 4.6 |
| LESS | $6N \cdot S \cdot V \cdot R + 2N \cdot V \cdot d \cdot R$ | 18 |

---

**Algorithm 1** DITS-iSFT-DPO

---

**Require:** Initial model $\theta_{\text{init}}$, problem Set $\mathcal{D}$, validation Set $\mathcal{D}_{\text{val}}$, and max iterations $T$
**Ensure:** parameter $\theta_T$
1: $\theta_0 \leftarrow \theta_{\text{init}}$
2: **for** $t = 1$ to $T$ **do**
3: $\quad D_t^{SFT} \leftarrow \text{SFTDataCollect}(\theta_{t-1})$ $\qquad\qquad\qquad\qquad$ ▷ Following Chen et al. (2024b)
4: $\quad \theta_t \leftarrow \text{SFT}(D_t^{SFT}, \theta_{\text{init}})$ $\qquad\qquad\qquad\qquad\qquad$ ▷ Following Chen et al. (2024b)
5: $\quad \mathcal{D}_t^{\text{DPO}} \leftarrow \emptyset$
6: $\quad$ **for all** $p_i \in \mathcal{D}$ **do**
7: $\qquad \mathcal{D}_i^{\text{DPO}} \leftarrow \text{MCTSSynthesis}(\theta_t, p_i)$
8: $\qquad \mathcal{I}_{\mathcal{F}_{\text{val}}} \leftarrow \text{DataInfluenceCollect}(\mathcal{D}_{\text{val}})$
9: $\qquad \mathcal{D}_t^{\text{DPO}} \leftarrow \mathcal{D}_t^{\text{DPO}} \cup \mathcal{D}_i^{\text{DPO}}$
10: $\quad$ **end for**
11: $\quad \mathcal{D}_t^{DPO} \leftarrow \text{InfluSelection}(\mathcal{D}_t^{DPO}, \mathcal{I}_{\mathcal{F}_{\text{val}}})$
12: $\quad \theta_t \leftarrow \text{DPO}(\mathcal{D}_t^{DPO}, \theta_t)$
13: **end for**
14: **return** $\theta_T$

---

# C ALGORITHM

The DITS-iSFT-DPO algorithm (Algorithm 1) iteratively refines a model by alternating between SFT and DPO. In each iteration, the model is fine-tuned using new data collected via SFT. Then, DPO training data is generated through MCTS synthesis and filtered using influence scores from a validation set. The model is updated with DPO to better align with preferences, leading to progressive improvement.

# D METHOD DETAILS

## D.1 REWARD FUNCTION

Following Optima (Chen et al., 2024b), we define each trajectory $\tau_i$ is then evaluated using a reward function $R : \mathcal{T} \to \mathbb{R}$:

$$R(\tau_i^j) = R_{\text{task}}(\tau_i^j) - \lambda_{\text{token}} R_{\text{token}}(\tau_i^j) + \lambda_{\text{loss}} \frac{1}{R_{\text{loss}}(\tau_i^j)}. \tag{15}$$

Here, $R_{\text{task}} : \mathcal{T} \to \mathbb{R}$ is the task-specific performance metric, $R_{\text{token}}(\tau_i^j) = \frac{\#\text{Tokens}(\tau_i^j)}{\max_k(\{\#\text{Tokens}(\tau_i^k)\}_k)}$ is the normalized token count, and $R_{\text{loss}}(\tau_i^j) = g\big(\mathcal{L}(\mathcal{M}_{\text{base}}, d_i, \tau_i^j)\big)$ is based on the language modeling loss of the base model $\mathcal{M}_{\text{base}}$. The positive coefficients $\lambda_{\text{token}}$ and $\lambda_{\text{loss}}$ are hyper-parameters. More details can refer to Optima (Chen et al., 2024b).

## D.2 INITIAL DATA FILTERING

For the preference data pairs obtained from the MCTS tree, we follow the Optima (Chen et al., 2024b) by initially filtering the data pair $(s, a_i^h, a_i^l)$. Specifically, we select pairs that satisfy: (1) $R(s, a_i^h) > \lambda_{\text{dpo-filter}}$. (2) $R(s, a_i^h) - R(s, a_i^l) > \lambda_{\text{dpo-diff}}$. (3) For preference pairs starting with the same problem $p$, we rank these pairs based on their Q-values and select the top 50% of the pairs.

|  | HotpotQA | 2WMH QA | TrivalQA | CBT | ARC-C | MMLU |
|---|---|---|---|---|---|---|
| **_DITS-iSFT-DPO_** | | | | | | |
| $\gamma$ | 1 | 1 | 1 | 1 | 1 | 1 |
| $\alpha$ | 0.5 | 0.5 | 0.5 | 0.5 | 0.5 | 0.5 |
| SFT LR | 2e-5 | 2e-5 | 2e-5 | 2e-5 | 1e-6 | 1e-6 |
| SFT Epoch | 2 | 1 | 1 | 1 | 4 | 2 |
| SFT Batch Size | 32 | 32 | 32 | 32 | 16 | 16 |
| $\lambda_{token}$ | 0.6 | 0.6 | 0.6 | 0.6 | 0.5 | 0.6 |
| $\lambda_{loss}$ | 1 | 1 | 1 | 1 | 0.6 | 0.7 |
| $\lambda_{\text{dpo-filter}}$ | 0.4 | 0.4 | 0.4 | 0.4 | 0.45 | 0.4 |
| $\lambda_{\text{dpo-diff}}$ | 0.2 | 0.2 | 0.2 | 0.2 | 0.2 | 0.2 |
| _Iteration 0_ | | | | | | |
| DPO LR | 5e-7 | 5e-7 | 5e-7 | 5e-7 | 5e-7 | 5e-7 |
| DPO Epoch | 1 | 1 | 1 | 1 | 1 | 1 |
| DPO Batch Size | 64 | 64 | 64 | 64 | 64 | 64 |
| $\beta$ | 0.5 | 0.5 | 0.7 | 0.7 | 0.1 | 0.1 |
| _Iteration 1_ | | | | | | |
| DPO LR | 5e-7 | 5e-7 | 5e-7 | 5e-7 | 5e-7 | 5e-7 |
| DPO Epoch | 1 | 1 | 1 | 1 | 1 | 1 |
| DPO Batch Size | 64 | 64 | 64 | 64 | 64 | 64 |
| $\beta$ | 0.5 | 0.5 | 0.7 | 0.7 | 0.1 | 0.1 |
| _Iteration 2_ | | | | | | |
| DPO LR | 5e-7 | 5e-7 | 5e-7 | 5e-7 | 5e-7 | 1e-6 |
| DPO Epoch | 1 | 1 | 1 | 1 | 1 | 1 |
| DPO Batch Size | 64 | 64 | 64 | 64 | 64 | 64 |
| $\beta$ | 0.5 | 0.5 | 0.7 | 0.5 | 0.2 | 0.1 |

Table 6: Hyper-parameters used in Table 1.

|  | HotpotQA | 2WMH QA | TrivalQA | CBT | ARC-C | MMLU |
|---|---|---|---|---|---|---|
| **_DITS-DPO_** | | | | | | |
| $\gamma$ | 1 | 1 | 1 | 1 | 1 | 1 |
| $\alpha$ | 0.5 | 0.5 | 0.5 | 0.5 | 0.5 | 0.5 |
| $\lambda_{token}$ | 0.6 | 0.6 | 0.6 | 0.6 | 0.5 | 0.6 |
| $\lambda_{loss}$ | 1 | 1 | 1 | 1 | 0.6 | 0.7 |
| $\lambda_{\text{dpo-filter}}$ | 0.4 | 0.4 | 0.4 | 0.4 | 0.45 | 0.4 |
| $\lambda_{\text{dpo-diff}}$ | 0.2 | 0.2 | 0.2 | 0.2 | 0.2 | 0.2 |
| DPO LR | 5e-6 | 5e-7 | 5e-6 | 5e-7 | 5e-7 | 5e-7 |
| DPO Epoch | 1 | 1 | 1 | 1 | 1 | 1 |
| DPO Batch Size | 64 | 64 | 64 | 64 | 64 | 64 |
| $\beta$ | 0.5 | 0.5 | 0.7 | 0.5 | 0.4 | 0.1 |

Table 7: Hyper-parameters used in Table 3.

## E  TRAINING DETAILS

The hyperparameters we used are shown in Table 6 and Table 7.

## F  DISTRIBUTION ANALYSIS

We visualize the distributions of Q-values and influence scores on the CBT, TrivalQA, and ARC-C datasets in Figure 4, highlighting the distribution of the top 30% data points selected by our methods with $\gamma = 1$. From the figures, we observe the following: (1) There are discrepancies between the influence score and the Q-value, indicating that the Q-value is not perfectly aligned with training needs. This underscores the importance of incorporating the influence score into both the MCTS

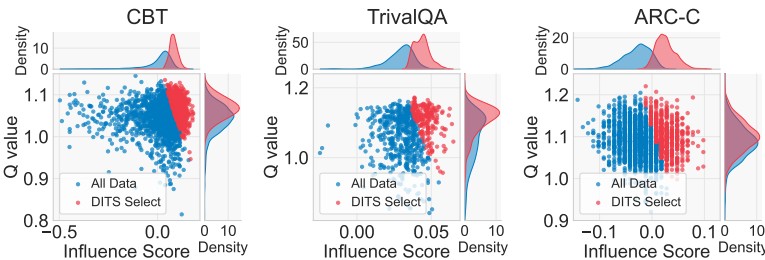

Figure 4: The scatter plot and density plots of Q-values and influence scores for synthetic data. The top 30% of the data selected by DITS is highlighted in red.

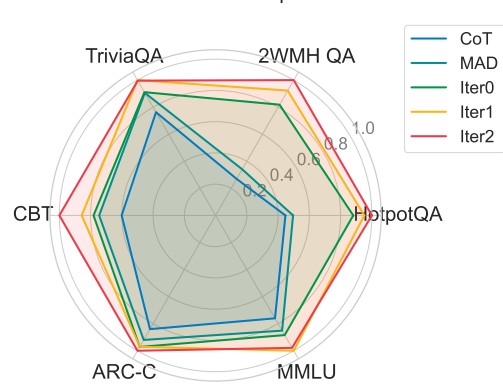

Figure 5: The relative performance improvement of DITS-iSFT-DPO across all datasets at different iterations. The best performance of each dataset is set as 1.0.

process and the data selection strategy. (2) The data selected by our method exhibit both high influence scores and Q-values, demonstrating that DITS effectively identifies and selects high-quality data. (3) In the mathematics dataset, the distribution of the influence score is concentrated at several discrete points. This is because the dataset is relatively challenging, and significant model improvements may be required to change the correctness of certain answers. As a result, the metric exhibits a stepwise effect.

## G  ABLATION STUDY

### G.1  ITERATIVE TIMES ANALYSIS

Using the performance of CoT as the baseline, we report the average relative performance improvement of our method, DITS-iSFT-DPO, across all datasets per iteration and present the results in Figure 5. From the figure, we observe that: (1) Our method achieves an average improvement of 91% compared to the single-agent CoT approach and an improvement of 64% over the multi-agent MAD method, demonstrating the effectiveness of our approach. (2) As the number of iterations increases, the average performance continues to improve. Since we start training from the same initial model in each iteration, this indicates that training better models and subsequently synthesizing data can consistently enhance the quality of the generated data and improve the final performance.

### G.2  VALIDATION SET SIZE ANALYSIS

To investigate the relationship between DITS and validation set size, we conducted additional experiments on 2WMHQA. As shown in the Figure 6, the performance of DITS-DPO improves with an increasing validation set size, indicating that a larger validation set provides a more accurate estimation. However, since a larger validation set demands a higher synthesis budget, a trade-off is necessary in practical applications.

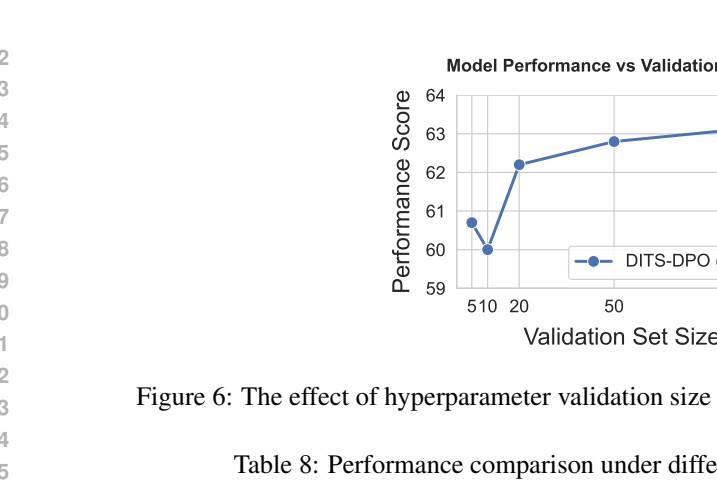

Figure 6: The effect of hyperparameter validation size $V$ on the performance of DITS.

Table 8: Performance comparison under different selection strategy.

| Model | 2WMH QA (V=20) | 2WMH QA (V=5) |
|---|---|---|
| Optima-DPO-Random-Select | 60.6 | 60.6 |
| DITS-DPO-High-Qvalue-High-Influence | 61.5 | 61.4 |
| DITS-DPO-Low-Qvalue-High-Influence | 61.7 | 60.8 |
| DITS-DPO-High-Qvalue-Low-Influence | 59.8 | 60.4 |

### G.3 SELECTION STRATEGY ANALYSIS

The influence scores are estimated on the validation set according to Eq equation 13. A larger size of validation set V yields a more accurate estimation but requires a higher synthesis budget.

1. Data points with high influence scores but low Q-values can still contribute to model training, and we have provided an example in the Appendix H. Empirical results, as shown in the Table 8 (V= 20), further validate this conclusion. However, when the estimated influence scores contain noise (due to a small V), supplementing the selection with high Q-values can help filter higher-quality data.

2. Data points with high Q-values but low influence scores represent data that the model already handles well. Further training on them risks overfitting and hinders model advancement. Empirically, these data points significantly degrade model performance, performing even worse than random select.

Overall, the contribution of data points to model training fundamentally depends on the influence scores. The correlation coefficient between Q-values and influence scores is approximately 0.1, indicating that traditional methods relying solely on high Q-values are insufficient for selecting optimal training data. When constrained by a limited synthesis budget—leading to less accurate influence score estimation—leveraging high Q-values as an auxiliary filtering criterion can improve data selection.

### G.4 RESULTS WITH DIFFERENT BASE MODELS

In this section, we evaluate DITS with Llama-3.1-8B-Instruct (Dubey et al., 2024) and Qwen2.5-7B-Instruct (Team, 2025) across information exchange and debate tasks. As shown in Table 9, we observe that across most datasets, DITS outperforms all baseline methods, demonstrating strong generalization and robustness. Moreover, experiments show that incorporating more powerful base models and applying task-specific fine-tuning further enhances performance, suggesting that stronger base models promote more effective multi-agent cooperation.

## H CASE STUDY

As illustrated in Figure 10, we present a comparative case study highlighting the differences in data selection outcomes when using Q-value versus influence score for the same task. The task—"Which

Table 9: **Results with different base models across Information Exchange and Debate tasks.** Best results are indicated in **bold**, and second-best results are underlined.

| Method | HotpotQA | 2WMH QA | CBT | ARC-C | MMLU |
|---|---|---|---|---|---|
| Qwen2.5-7B-Instruct | | | | | |
| - Base | 39.02 | 37.92 | 25.77 | 4.93 | 11.40 |
| - Optima-SFT | 49.08 | 58.18 | 52.63 | 73.98 | **57.70** |
| - Optima-DPO | **52.19** | 60.86 | 61.67 | 74.05 | 55.70 |
| - DITS-DPO | 51.68 | **61.81** | **62.23** | **74.23** | 56.30 |
| Llama-3.1-8B-Instruct | | | | | |
| - Base | 38.07 | 35.32 | 28.27 | 18.17 | 16.20 |
| - Optima-SFT | 44.60 | 38.25 | 53.79 | 68.26 | 48.10 |
| - Optima-DPO | 44.47 | 40.12 | 54.29 | 74.15 | 56.20 |
| - DITS-DPO | **45.23** | **41.22** | **61.44** | **76.37** | **58.00** |

film has the director who was born later, Eyes of the Forest or Stardust on the Sage?"—was analyzed through agent dialogues between Alice and Bob.

In the Q-value-selected data pair, the dialogue history efficiently conveyed the directors' birth dates within a single interaction round. The chosen response directly identified "Stardust on the Sage" as the correct answer using special token markers in the response, achieving an exceptionally high Q-value. Meanwhile, the rejected response, although redundant in restating first-round information, contained no errors, thereby maintaining a high Q-value. However, the minimal difference between the paired responses resulted in low influence scores, limiting their utility for model improvement.

In contrast, the influence-score-selected data pair exhibited incomplete information sharing in the dialogue history. The chosen response correctly ruled out Hillyer as the director of "Stardust on the Sage" but required more information to get to the correct answer, leading to lower Q-values. More critically, the rejected response contained hallucinatory content—an outright factual error falsely attributing Katedza as the film's director—which fundamentally obstructed correct reasoning and resulted in an extremely low Q-value. This high-contrast pair data holds significant pedagogical value, as it juxtaposes valid reasoning with critical hallucinations, thereby achieving superior influence scores.

Our analysis reveals that while high-Q-value pairs ensure accurate answers, they may have low influence scores and contribute little to multi-agent training. Conversely, data pairs with pronounced contrasts in reasoning validity—despite both exhibiting lower Q-values—substantially enhance model robustness against hallucinations by explicitly demarcating errors. These findings strongly advocate prioritizing influence score metrics over Q-value in both data synthesis and tree search to maximize model performance.

Table 10: Case study to demonstrate the data selected by Q-value and influence score on 2WMH QA dataset.

| Question | Which film has the director who was born later, Eyes Of The Forest or Stardust On The Sage? |
|---|---|
| **Q-value Select** | ```
Alice:
[
    ["""Film""", """Eyes of the Forest"""],
    ["""Director""", """Lambert Hillyer"""],
    ["""Birth Date""", """July 8, 1893"""],
    ["""Death Date""", """July 5, 1969"""],
    ["""Director Born Later?""", """0"""]
]
Bob:
[
    ["""Film""", """Stardust On The Sage"""],
    ["""Director""", """William Morgan"""],
    ["""Birth Date""", """1895-12-07"""],
    ["""Director Born Later?""", """1"""]
]
Chosen:
Alice:
[
    ["""Director Born Later?""", """1"""],
    ["""Conclusion""", """<A> Stardust On The Sage</A>"""]
]
Rejected:
Alice:
[
    ["""Director""", """Lambert Hillyer"""],
    ["""Birth Date""", """July 8, 1893"""],
    ["""Director Born Later?""", """0"""]
]
``` |
| **Influence Score Select** | ```
Alice:
[
    ["""Information""", """Lambert Harwood Hillyer was born 1893"""],
    ["""Query""", """Is Hillyer the director of Eyes Of The Forest?"""],
    ["""Information""", """Yes"""]
]
Bob:
[
    ["""Query""", """Is Hillyer the director of Stardust On The Sage?"""],
    ["""Information""", """No"""]
]
Chosen:
Alice:
[
    ["""Query""", """Is Hillyer the director of Stardust On The Sage?"""],
    ["""Information""", """No"""]
]
Rejected:
Alice:
[
    ["""Query""", """Who is the director of Stardust On The Sage?"""],
    ["""Information""", """Rumbi Katedza"""],
    ["""Query""", """Is Katedza the director of Stardust On The Sage?"""],
    ["""Information""", """Yes"""]
]
``` |