# OpenReview forum: "Efficient Multi-Agent System Training with Data Influence-Oriented Tree Search"
_ICLR.cc/2026/Conference — ICLR 2026 Conference Withdrawn Submission_

### Official Review · Reviewer_G333 · 2025-10-27

**Soundness:** 3
**Presentation:** 3
**Contribution:** 3
**Rating:** 6
**Confidence:** 3

**Summary:**

This paper introduces Data Influence-oriented Tree Search (DITS), a novel framework for generating more effective synthetic data to train multi-agent systems (MAS). The authors argue that current methods, which rely on Q-values to select data, are flawed because high Q-value data does not always lead to the best model performance. DITS solves this by using a novel influence score to guide the tree search, prioritizing data based on its expected contribution to the model's actual performance improvement rather than its Q-value. The paper also presents a computationally efficient method for estimating this influence score based on non-differentiable metrics. Experiments across seven datasets show that DITS achieves state-of-the-art performance and scales more efficiently than traditional approaches.

**Strengths:**

1. This work proposes a new, influence-aware framework that directly optimizes for data *utility* in training. This "influence score" is a more relevant metric for data synthesis than traditional Q-values.
2. A major strength is the new, computationally cheap method for estimating the influence score. By avoiding complex gradient computations and focusing on non-differentiable metrics, it scales more efficiently and reduces computational overhead.
3. The method achieves state-of-the-art results, demonstrating clear, quantitative improvements (e.g., 2.5%-2.7% performance gains) across seven datasets and three different multi-agent tasks.

**Weaknesses:**

1. The proposed method introduces an influence score metric to quantify the impact of data on the current agent’s performance. In previous methods, this score was developed to measure the difference in loss when a data point was assigned a higher weight in the training dataset. In contrast, this work adjusts the influence score based on changes in non-differentiable performance metrics. However, the work lacks a clear intuition to explain why this change enables the influence score to indicate performance improvement. Additionally, it is unclear why the authors believe that higher Q-values do not necessarily mean performance improvement.

2. The authors claim that training effective MAS requires high-quality data that reflects complex agent interactions, noting that it is costly and time-consuming to collect this data in the real world. They utilize MCTS to simulate interactions and automatically produce preference-labeled training data. In this process, the authors select a node for the candidate set using a softmax distribution of Q-values, thinking that the softmax distribution balances exploration and exploitation. However, in practice, the softmax distribution typically emphasizes exploitation in reinforcement learning. It does not inherently balance exploration and exploitation, which raises concerns about the proposed method's exploration capabilities.

3. The authors assume that the transition function is known during the data generation process. However, in complex multi-agent systems, the transition function is often unknown. This raises two questions: Can the proposed method work properly in model-free environments? Or, can we learn an accurate transition function from existing datasets?

**Questions:**

Please see the Weaknesses above.

---

> ### Author Response · Authors · 2025-11-21
> **Reply**
>
> Thanks for your constructive review. We are pleased that you find our influence-aware framework novel and recognize the practical advantages of our computationally efficient influence score estimation. We appreciate your positive assessment of the quantitative improvements achieved across multiple datasets and multi-agent tasks. Below, we address your comments point by point.
>
> >Q1: The work lacks a clear intuition to explain why this change enables the influence score to indicate performance improvement. Additionally, it is unclear why the authors believe that higher Q-values do not necessarily mean performance improvement.
>
> **AQ1:** Thanks for pointing this out. Our new metric is defined to measure the difference in non-differentiable performance metrics when a data point is added to the training set with a small weight. **As derived in Appendix A, selecting the most influential data points in this sense maximizes validation performance.** Regarding Q-values, our conclusion is primarily empirical: the data influence score serves as a gold standard for assessing a data point’s contribution to model improvement, and our experiments show that Q-values correlate poorly with influence scores, indicating that higher Q-values do not necessarily imply performance gains.
>
> >Q2: The softmax distribution typically emphasizes exploitation in reinforcement learning. It does not inherently balance exploration and exploitation, which raises concerns about the proposed method's exploration capabilities.
>
> **AQ2:** Thanks for raising this concern. When using softmax sampling with a relatively high temperature, the selection approximates random node expansion, introducing sufficient exploration. Additionally, we filter out nodes that are similar to already explored ones, further enhancing the diversity and exploration of the search.
>
>
> >Q3: The authors assume that the transition function is known during the data generation process. However, in complex multi-agent systems, the transition function is often unknown. This raises two questions: Can the proposed method work properly in model-free environments? Or, can we learn an accurate transition function from existing datasets?
>
> **AQ3:** Thanks for raising this concern. We would like to clarify that we are not entirely sure whether your reference to the transition function concerns the interactions between agents or the environment’s state-transition probabilities. For agent–agent interactions, we have validated the effectiveness of our method in dynamic scenarios. Regarding the environment’s state-transition function, we believe it can be obtained through direct interaction with the real environment during data synthesis.

---

### Official Review · Reviewer_gPdX · 2025-10-31

**Soundness:** 3
**Presentation:** 2
**Contribution:** 2
**Rating:** 4
**Confidence:** 3

**Summary:**

This paper proposes DITS, a data influence–oriented tree search framework for multi-agent synthetic data generation and self-training. Rather than relying on MCTS Q-values alone, it estimates each sample’s influence on downstream validation metrics (including non-differentiable ones) via a one-step perturbation/finite-difference approximation and uses this to guide both tree expansion and preference selection with H(z)=IF_val(z)+ $\gamma$Q. Experiments on static and dynamic multi-agent tasks show consistent improvements over Optima-style baselines under comparable or lower compute budgets.

**Strengths:**

- Originality and clarity: The paper reframes multi-agent data selection around validation-aligned influence, integrating a finite-difference influence estimator (applicable to non-differentiable metrics) directly into tree search and preference construction; the objective and algorithm are clearly specified with equations and a concise pipeline.
- Quality and significance: Across static and dynamic multi-agent settings, the method consistently improves over strong Optima-style baselines with modest computational overhead; ablations and budget analyses suggest the approach is practical and broadly applicable to MAS data synthesis and self-training.

**Weaknesses:**

- Heuristic design choices: Many key components are tuned heuristically without principled justification or sensitivity studies—e.g., the edit-distance threshold for candidate pruning, the per-round selection ratio α, Softmax sampling solely on Q (instead of UCB/PUCT), and a Q-based prefilter before the final ranking. Several rewards/regularizers are also combined with ad-hoc weights, making behavior sensitive to undocumented scales.
- Linear mixing without calibration: The scoring rule $H(z)=IF_{val}(z)+\gamma Q$ lacks scale alignment and any adaptive weighting. Since $IF_{val}$ and $Q$ can live on different (and drifting) ranges across rounds/tasks, a fixed $\gamma$ risks unstable selection; no normalization (e.g., z-score/tempering) or learned mixture is explored.
- Crude influence approximation: The estimator relies on a one-step perturbation and finite differences on non-differentiable metrics, which can be highly noisy and step-size dependent in non-convex DPO training. Using LoRA gradients to proxy full-parameter updates further increases mismatch between estimated and realized influence.
- Tiny and reused validation set: A very small validation split (V=20) is reused to compute $IF_{val}$ across iterations, inviting high variance, overfitting to the validation set, and selection leakage; there is no cross-folding, multiple seeds, or reporting of uncertainty to mitigate these risks.

**Questions:**

- Calibration and selection policy: Have you explored normalization or adaptive weighting for the scoring rule $H(z)=IF_{val}(z)+\gamma Q$ (e.g., per-round z-scoring, temperature scaling, or a learned mixture), and ablations that remove the Q-based prefilter or replace Softmax-on-Q with UCB/PUCT? How do these choices affect stability and diversity of selected data?

- Robustness of influence estimation: How sensitive are results to the step-size/perturbation choices, the one-step vs multi-step approximation, LoRA vs full-parameter gradients, and the small reused validation set (V=20)? Can you report seed-averaged curves and show that estimated influence correlates with realized improvement on a held-out set or under cross-fold validation?

---

> ### Author Response · Authors · 2025-11-21
> **Reply (1)**
>
> Thanks for your constructive review. We are pleased that you find our reformulation of multi-agent data selection to be original and clearly presented, and that you recognize the practical significance of our influence-based approach in both static and dynamic MAS settings. We appreciate your positive assessment of the method’s empirical gains, computational efficiency, and broad applicability. Below, we address your concerns point by point.
>
> >W1: Several design choices rely on ad-hoc heuristics without principled motivation or sensitivity analysis.
>
> **A1:** Thanks for raising this concern. Regarding the design choices in the MCTS procedure, our primary goal was to remain consistent with the Optima baseline in order to ensure a fair comparison. We agree that these components can influence the quality of the synthesized data; however, under different reasonable design choices, our method consistently identifies data that is more beneficial for model training.
>
> >W2: Linear mixing without calibration
>
> **A2:** Thank you for pointing out the issue of scale alignment in the linear mixing rule. In fact, we do apply normalization to $IF_{val}$ using a min–max scheme, $(x−min)/(max−min)$. Since the Q-values are inherently bounded within
> $[0,1]$, additional normalization is unnecessary in practice. We will revise the manuscript to make this normalization step explicit.
>
> >W3: The estimator relies on a one-step perturbation and finite differences on non-differentiable metrics, which can be highly noisy and step-size dependent in non-convex DPO training. Using LoRA gradients to proxy full-parameter updates further increases mismatch between estimated and realized influence.
>
> **A3:** Thanks for raising this concern. We acknowledge that the one-step perturbation and finite-difference estimation on non-differentiable metrics can introduce bias. This is indeed a trade-off we make to substantially reduce computational overhead. With additional computational resources, one could incorporate more update steps or higher-fidelity gradient approximations to obtain more accurate data influence estimation.
>
> >W4: Tiny and reused validation set: A very small validation split (V=20) is reused to compute across iterations, inviting high variance, overfitting to the validation set, and selection leakage; there is no cross-folding, multiple seeds, or reporting of uncertainty to mitigate these risks.
>
> **A4:** Thanks for raising this concern. To show the robustness of validation set, we conduct two additional experiments.
>
> 1. We conduct an analytical experiment: randomly selecting 20 samples as a validation set and 200 samples as another validation set. We then compute the influence score for each data point and calculate the correlation between these two sets of scores. **The resulting correlation coefficient is 0.912, indicating a strong correlation and demonstrating that the randomly selected 20 samples already exhibit good representativeness.**
> 2. Additionally, we perform supplementary experiments on the 2WMH QA dataset, where we **select different validation sets (Size:20)** to compute the influence scores of data points. Based on these scores, we train models and obtained the results as shown in the table.
>
> | model | Validation Set 1 | Validation Set 2 |Validation Set 3 | Validation Set 4 | Validation Set 5 |
> | -------- | -------- | -------- |-------- | -------- | -------- |
> | DITS-DPO     | 62.2     | 61.8    | 62.0 | 62.4 | 61.9 |
>
> As shown in the table, the proposed method exhibits strong robustness with respect to different validation sets, **maintaining consistently low variance. Employing larger validation sets can further reduce the variance.**

---

> > ### Author Response · Authors · 2025-11-21
> > **Reply (2)**
> >
> > >Q1: Calibration and selection policy: Have you explored normalization or adaptive weighting for the scoring rule
> >  (e.g., per-round z-scoring, temperature scaling, or a learned mixture), and ablations that remove the Q-based prefilter or replace Softmax-on-Q with UCB/PUCT? How do these choices affect stability and diversity of selected data?
> >
> > **AQ1:**
> > 1. As discussed in A2, we applied min-max normalization in our methods. To validate other calibration methods, we conducted additional experiments on HotpotQA dataset. We observed that, **compared with other calibration strategies, min–max normalization performs best in our setting** and empirically produces a distribution of data influence scores that aligns most compatibly with the Q-values for mixture and ranking.
> >
> > | Method | Per round Min-Max (our method) | Per round z-scoring | Temperature scaling |
> > | ---- |----| ---| ---|
> > |  DITS-DPO  |  **52.8** |  52.1  |  51.8 |
> >
> >
> > Prior to performing ablations on the data generation pipeline, **it is worth noting that modifying the data generation pipeline will alter the quality of the synthesized dataset.** The resulting performance degradation will largely be attributed to the inability to obtain high-quality data under alternative generation schemes. In the  paper, we did not conduct ablations on the data synthesis process in order to preserve fairness in comparison with prior baselines.
> >
> > 2. We conducted an ablation in which the Q-based prefilter was removed, and we found that **incorporating our influence-based method enhances the robustness of the selection process even without this prefilter.**
> >
> > | Method |   Optima-DPO w/ Q-based prefilter | Optima-DPO w/o Q-based prefilter | DITS-DPO w/ Q-based prefilter | DITS-DPO w/o Q-based prefilter |
> > | ----| --- | --- | --- | --- |
> > | HotpotQA |  46.6 |  35.2 |  52.8 |  50.2 |
> >
> >
> > 3. Classical UCB and PUCT select expansion nodes within the same tree depth. However, due to the high computational cost of node expansion in LLM-based settings, **we follow Optima’s strategy and allow node selection across different depths.** This design ensures that even with a relatively small number of rollouts, the search can still produce sufficiently diverse generated data. Because of our limited computational budget, we were unable to perform a thorough evaluation of the effectiveness of standard UCB and PUCT strategies in this setting.
> >
> >
> > >Q2: Robustness of influence estimation: How sensitive are results to the step-size/perturbation choices, the one-step vs multi-step approximation, LoRA vs full-parameter gradients, and the small reused validation set (V=20)? Can you report seed-averaged curves and show that estimated influence correlates with realized improvement on a held-out set or under cross-fold validation?
> >
> > **AQ2:**
> >
> > 1. From a theoretical standpoint, a single gradient update already provides a first-order influence approximation, and taking multiple steps does not asymptotically lead to a more accurate estimate. Empirically, we observe that using a small number of steps can offer minor improvements over the one-step estimate, whereas larger step counts introduce significant bias and ultimately harm model performance.
> >
> > | Method | DITS-DPO (step: 1) | DITS-DPO (step:2) | DITS-DPO (step: 5) |
> > | ---- | ----| ---| ---|
> > | HotpotQA | 52.8 | 53.2 | 51.5 |
> >
> > 2. The computational and memory costs of full-parameter fine-tuning are prohibitively high, and we currently lack the resources to conduct such an evaluation.
> >
> > 3. We constructed multiple validation sets on 2WMH QA using different random seeds to assess the robustness of our approach.
> >
> > | model | Validation Set 1 | Validation Set 2 |Validation Set 3 | Validation Set 4 | Validation Set 5 |
> > | -------- | -------- | -------- |-------- | -------- | -------- |
> > | DITS-DPO     | 62.2     | 61.8    | 62.0 | 62.4 | 61.9 |
> >
> > As shown in the table, the proposed method exhibits strong robustness with respect to different validation sets, **maintaining consistently low variance. Employing larger validation sets can further reduce the variance.**

---

> > ### Comment · Reviewer_gPdX · 2025-11-27
> >
> > The authors' rebuttal addresses some of my concerns, and I have decided to maintain the original score.

---

### Official Review · Reviewer_LZwz · 2025-11-01

**Soundness:** 3
**Presentation:** 3
**Contribution:** 2
**Rating:** 4
**Confidence:** 3

**Summary:**

The paper introduces DITS, a training framework that leverages influence scores to guide data selection, aiming to address the mismatch between Q-value-based estimation and the model’s training objective in prior methods. The approach is evaluated across multiple multi-agent tasks, demonstrating its effectiveness.

**Strengths:**

The paper is clearly written. The empirical section presents comparative experiments in both static and dynamic scenarios, and includes extensive hyperparameter-control experiments that demonstrate the method’s robustness.

**Weaknesses:**

1. The data selection strategy yields modest gains in downstream performance but introduces additional LoRA training overhead. It also relies on a validation set whose distribution closely aligns with the evaluation set, and its impact on generalization in multi-agent systems is not evaluated.
2. Some citations need format correction. When citing authors after mentioning a method name, the citation should be enclosed in parentheses (use `\citep` rather than `\citet`).

**Questions:**

1. In the dynamic scenario experiments (Table 2), why are results under the Optima-iSFT-DPO and DITS-iSFT-DPO settings not reported?
2. My understanding is that DITS builds on Optima by further selecting training data using a validation set from the same distribution. My main concern is that DITS yields modest gains and lacks transfer evaluation (training on one dataset and testing on another within the same domain), so the effect of the data selection strategy on generalization remains unvalidated.
3. In Optima, the Q-value used for trajectory selection already includes $R_{task}$, which directly reflects downstream task performance. In DITS, changes in validation-set performance are again used for further data selection. Is the misalignment between the Q-value and the influence score (as mentioned in Table 1) primarily due to distribution differences between the training and validation sets, or to the additional terms introduced into the Q-value ($R_{token}$, $R_{loss}$ in Equation 15), which appear to trade off task performance, token cost, and linguistic fluency?

---

> ### Author Response · Authors · 2025-11-21
> **Reply (1)**
>
> Thanks for your instructive review. We are glad that you regard our paper as clearly written and our method as highly robust. We will address your concern point by point.
>
> >w1: The data selection strategy yields modest gains in downstream performance but introduces additional LoRA training overhead. It also relies on a validation set whose distribution closely aligns with the evaluation set, and its impact on generalization in multi-agent systems is not evaluated.
>
> **A1:** Thanks for raising this concern. As we discussed in Figure 1(d) and table 4, we have demonstrated that, under the same computational budgets, our method achieves superior empirical performance relative to Optima. More importantly, our key insight is that, under constrained computational resources, allocating a portion of the budget to influence–score estimation can indeed yield greater performance gains.
>
> To validate the generalizability of our approach, we evaluate models trained on the HotpotQA dataset on two out-of-domain benchmarks: 2WMH QA and TriviaQA.
>
> | method | MAD |    AutoForm |    DITS-iSFT-DPO (trained on Hotpot QA) | DITS-iSFT-DPO (trained on 2WMH QA) |
> | ------ | ------- | -------   | ----- | ----|
> | 2WMH QA  |  25.9   |   24.7  |  **58.4**    | 76.0 |
>
>
> | method | MAD |    AutoForm |    DITS-iSFT-DPO (trained on Hotpot QA) | DITS-iSFT-DPO (trained on TriviaQA) |
> | ------ | ------- | -------   | ----- | ----|
> | TriviaQA  |  71.0   |   60.9  |  **73.2**    | 78.4 |
>
> As shown in the table above, although the performance is not as strong as that of models fine-tuned directly on the target dataset, **our method achieves substantial improvements over the baselines.** This demonstrates that our validation-set–based data selection strategy exhibits strong generalization ability. Moreover, if we want to improve cross-dataset robustness in the future, the validation set can be constructed by mixing samples from multiple dataset sources.
>
>
>
> >w2: Some citations need format correction. When citing authors after mentioning a method name, the citation should be enclosed in parentheses (use \citep rather than \citet).
>
> **A2:** Thanks for pointing this out. We have revised the citation formats in the new version.
>
> >Q1: In the dynamic scenario experiments (Table 2), why are results under the Optima-iSFT-DPO and DITS-iSFT-DPO settings not reported?
>
> **AQ1:** Thanks for raising this concern. The core contribution of our method lies in selecting the most informative data for model training—specifically, using data influence as the criterion for data selection instead of Q-value. This idea can be effectively validated within a single iteration setting. Therefore, in the dynamic scenario, we report only single iteration experiments, as they are sufficient to demonstrate the validity of our approach.
>
> >Q2: The effect of the data selection strategy on generalization remains unvalidated.
>
> **AQ2:** Thanks for pointing this out. To validate the generalizability of our approach, we evaluate models trained on the HotpotQA dataset on two out-of-domain benchmarks: 2WMH QA and TriviaQA.
>
> | method | MAD |    AutoForm |    DITS-iSFT-DPO (trained on Hotpot QA) | DITS-iSFT-DPO (trained on 2WMH QA) |
> | ------ | ------- | -------   | ----- | ----|
> | 2WMH QA  |  25.9   |   24.7  |  **58.4**    | 76.0 |
>
>
> | method | MAD |    AutoForm |    DITS-iSFT-DPO (trained on Hotpot QA) | DITS-iSFT-DPO (trained on TriviaQA) |
> | ------ | ------- | -------   | ----- | ----|
> | TriviaQA  |  71.0   |   60.9  |  **73.2**    | 78.4 |
>
> As shown in the table above, although the performance is not as strong as that of models fine-tuned directly on the target dataset, **our method achieves substantial improvements over the baselines.** This demonstrates that our validation-set–based data selection strategy exhibits strong generalization ability. Moreover, if we want to improve cross-dataset robustness in the future, the validation set can be constructed by mixing samples from multiple dataset sources.

---

> > ### Author Response · Authors · 2025-11-21
> > **Reply (2)**
> >
> > >Q3: Is the misalignment between the Q-value and the influence score (as mentioned in Table 1) primarily due to distribution differences between the training and validation sets, or to the additional terms introduced into the Q-value (in Equation 15), which appear to trade off task performance, token cost, and linguistic fluency?
> >
> > **AQ3:** Thanks for raising this concern. We argue that the fundamental difference between Q-values and data influence scores arises from the distinct perspectives they use to assess data quality. Successful trajectory data does not necessarily improve model performance; in some cases, it may even degrade model generalization. In response to your concerns, we address the issue from two angles:
> >
> > 1. The validation set and training set are constructed by randomly sampling two non-overlapping subsets from the same dataset. Therefore, there is no distributional shift in a statistical sense.
> >
> > 2. We further evaluated the correlation between Q-values and data influence scores on the HotpotQA dataset under the setting where only $R_{task}$ is included. The correlation is 0.1547, which remains extremely low. This indicates that the discrepancy does not stem from a trade-off with other reward components.

---

> > > ### Comment · Reviewer_LZwz · 2025-11-27
> > >
> > > I appreciate the authors’ explanations and efforts, which have addressed part of my concerns. However, several issues remain:
> > >
> > > **Regarding A1 and AQ2**
> > > I appreciate the additional experiments conducted to validate the generalizability of DITS. However, since MAD and AutoForm are training-free methods, comparisons between DITS and the direct baseline Optima under the same training settings would make your claim more convincing. This would help demonstrate whether targeted optimization on downstream performance provides greater benefits than relying on Q-values.
> > >
> > > **Regarding AQ1**
> > > Thank you for the explanations. To clarify my original question: is there an absence of evaluation results for both Optima and DITS when they are trained in a non-iterative SFT-then-DPO setting in dynamic scenarios?
> > >
> > > **Other Suggestions**
> > > To clarify and strengthen the motivation of this work, the authors should consider moving parts of the content currently in the appendices—specifically, *Selection Strategy Analysis* and *Case Study*—into the main body of the paper.

---

### Note · Authors · 2026-01-03

I have read and agree with the venue's withdrawal policy on behalf of myself and my co-authors.